# Direct Plasmonic Solar Cell Efficiency Dependence on Spiro-OMeTAD Li-TFSI Content

**DOI:** 10.3390/nano11123329

**Published:** 2021-12-08

**Authors:** Xinjian Geng, Mohamed Abdellah, Robert Bericat Vadell, Matilda Folkenant, Tomas Edvinsson, Jacinto Sá

**Affiliations:** 1Department of Chemistry—Angstrom, Uppsala University, 751 20 Uppsala, Sweden; xinjian.geng@kemi.uu.se (X.G.); robert.bericat.vadell@kemi.uu.se (R.B.V.); 2R&D Division, Peafowl Solar Power AB, 756 43 Uppsala, Sweden; mo.abdellah@peafowlsolarpower.com (M.A.); matildafolkenant@gmail.com (M.F.); 3Department of Chemistry, Qena Faculty of Science, South Valley University, Qena 83523, Egypt; 4Department of Materials Science and Engineering—Solid State Physics, Uppsala University, 751 20 Uppsala, Sweden; tomas.edvinsson@angstrom.uu.se; 5Institute of Physical Chemistry, Polish Academy of Sciences (IChF-PAN), 01-224 Warsaw, Poland

**Keywords:** direct plasmonic solar cell, hole transporting material conductivity, ultrafast transient spectroscopy

## Abstract

The proliferation of the internet of things (IoT) and other low-power devices demands the development of energy harvesting solutions to alleviate IoT hardware dependence on single-use batteries, making their deployment more sustainable. The propagation of energy harvesting solutions is strongly associated with technical performance, cost and aesthetics, with the latter often being the driver of adoption. The general abundance of light in the vicinity of IoT devices under their main operation window enables the use of indoor and outdoor photovoltaics as energy harvesters. From those, highly transparent solar cells allow an increased possibility to place a sustainable power source close to the sensors without significant visual appearance. Herein, we report the effect of hole transport layer Li-TFSI dopant content on semi-transparent, direct plasmonic solar cells (DPSC) with a transparency of more than 80% in the 450–800 nm region. The findings revealed that the amount of oxidized spiro-OMeTAD (spiro^+^TFSI^−^) significantly modulates the transparency, effective conductance and conditions of device performance, with an optimal performance reached at around 33% relative concentration of Li-TFSI concerning spiro-OMeTAD. The Li-TFSI content did not affect the immediate charge extraction, as revealed by an analysis of electron–phonon lifetime. Hot electrons and holes were injected into the respective layers within 150 fs, suggesting simultaneous injection, as supported by the absence of hysteresis in the I–V curves. The spiro-OMeTAD layer reduces the Au nanoparticles’ reflection/backscattering, which improves the overall cell transparency. The results show that the system can be made highly transparent by precise tuning of the doping level of the spiro-OMeTAD layer with retained plasmonics, large optical cross-sections and the ultrathin nature of the devices.

## 1. Introduction

In 2019, the US Energy Information Administration (EIA) projected that world energy consumption would grow by nearly 50% between 2018 and 2050 [1], from which a large part is expected to derive from renewable sources. The COVID-19 pandemic reduced the global energy demand by 3.8% in the first quarter of 2020 [2]. Interestingly, renewable energy emerged as the most resilient energy source to COVID-19 lockdown measures, becoming the only energy source that posted growth in demand in Q1 2020 (an increase of about 1.5% relative to Q1 2019), driven by larger installed capacity and priority dispatch. 

Photovoltaics are a fundamental part of renewable electricity generation activities, which increased by almost 3% in Q1 2020. However, apart from supplying cost-effective energy on a large scale, the last decade saw increased usage of photovoltaics as energy harvesters for low-power devices such as the internet of things (IoT) sensors and e-paper displays. Currently, many IoT devices are powered by single-use batteries that have a limited lifetime and insufficient sustainability credentials. There is, therefore, a strong interest in self-powered devices or alternative energy sources to continuously power IoT devices [3], and photovoltaic power has emerged as a solid contender to power them [4]. The proliferation of energy harvesting solutions is strongly associated with technical performance, cost and aesthetics, with the latter often being the driver of adoption. Therefore, colored and highly transparent solar cells have become ever more relevant and sought after [5,6]. 

A plethora of photovoltaic technologies have emerged, from silicon solar cells and thin-film, solid-state semiconductors to dye-sensitized solar cells, organic photovoltaics and perovskites [7]. Direct plasmonic solar cells (DPSC) is an emergent but relatively unknown technology that was first reported in 2010 by Nishijima et al. [8]. Plasmonic materials have optical cross-sections that exceed their geometric size by tenfold, making them uniquely suitable for converting light into usable energy and carriers. DPSC are particularly ideal for applications requiring high levels of transparency and/or soft color shades. Despite being a relatively new technology, they are being explored for commercialization by Peafowl Solar Power AB, propelled by their unmatchable level of transparency. It is essential to note that in DPSC, plasmonic material is utilized as direct light absorbers and should not be confused with technologies where plasmonics are exploited to enhance light absorption.

Optical excitation of the localized surface plasmon creates hot carriers [9,10,11] that can be exploited to produce electricity [12,13]. The energies of the hot carriers remain incommensurate but are believed to be equal to the absorbed photon energy [14]; they rapidly undergo carrier multiplication resulting in a decrease in their average energy but an increase in their number as it is an elastic process [15]. The ultrafast charge relaxation and recombination [16] and the absence of an energy gap in these absorbing materials request the transference of the hot carriers to suitable acceptors/transport layers to expedite their lifetime [13,17,18], which, in the case of photovoltaics, also determines their open-circuit voltage (*V*_oc_). Therefore, to ensure stable *V*_oc_, commercial DPSC must always be constructed using an n–i–p or p–i–n architecture similar to organic and perovskite photovoltaics, as explained vide infra. The electronic affinity of the charge to the specific transporting layer is believed to be the driver of the transport [19].

DPSC architecture resembles organic and perovskite photovoltaic cell structures consisting of active material, in this case, plasmonic nanoparticles, sandwiched between electron- and hole-transporting materials. However, in DPSC, the active material layer is not compact, i.e., it is composed of sprinkled and completely separated nanoparticles, thus ensuring that each particle works as a single resonator. As a consequence, high transparency DPSC can be manufactured, in which a local field due to multi-particle resonance is prevented [20,21], but with compromising overall efficiency as only a tiny fraction of the light is absorbed.

Reineck et al. [22] reported solid-state DPSC consisting of a monolayer of active material (Ag or Au nanoparticles (NPs)) sandwiched between TiO_2_ and 2,2′,7,7′-tetrakis-(N,N-di-4-methoxyphenylamino)-9,9′-spirobifluorene (spiro-OMeTAD) that worked as electron- and hole-transporting materials, respectively. The cells had an average incident photon-to-electron conversion efficiency (IPCE) of 1.0 ± 0.2% (Au NPs) and 0.5 ± 0.35% (Ag NPs), with champion cells attaining IPCE values of 1.8% (Au) and 1.4% (Ag). The authors estimated that a monolayer of metal NPs absorbed up to 37% of the incident photons at the plasmon resonance wavelengths, yielding absorbed IPCE of about 4.9% for gold NPs and 3.8% for silver NPs for the champion cells. The average *V*_oc_ of the cells was in the region of 200 mV. A later report showed that quantum efficiency depends on particle size, with smaller sizes showing higher efficiencies [23]. Note that in both studies the authors used spiro-OMeTAD without tris(2-(1*H*-pyrazol-1-yl)-4-*tert*-butylpyridine)cobalt(III) bis(trifluoromethane)sulfonimide (FK209), making the material colorless and more transparent but less efficient, according to studies on perovskites solar cells [24,25].

The works of Reineck et al. [22,23] provided the foundation for solid-state DPSC that could eventually be processed using low-temperature manufacturing techniques, such as printing and coating. However, no investigations have been carried out to further our understanding of the technology and hopefully maximize its potential. Considering the DPSC architecture of Reineck et al. [22,23], there are two pressing, open questions: (i) Are the two charges injected simultaneously or singly? (ii) Is the oxidized content of spiro-OMeTAD a determinant of device performance? Our group has demonstrated that plasmonic hot electrons and holes can be injected into specific accepting layers individually and simultaneously [11,14,18]. However, the studies were not performed on photovoltaic architectures that have been proven to work. Oxidized spiro species are intimately related to their efficiency as hole transporting material [26]; however, if this is relevant to systems such as transparent DPSC where the photocurrent is comparatively low remains untested.

Herein, the two open questions regarding plasmonic solar cells are addressed. The DPSC consisted of a sub-monolayer of Au NPs on TiO_2_/FTO glass, and spiro-OMeTAD with different ratios of Li-TFSI and 4-*tert*-butylpyridine (tBP) was produced. To address the effect of oxidized spiro content on the overall efficiency, we varied the content of bis(trifluoromethane)sulfonimide lithium salt (Li-TFSI) in spiro-OMeTAD on DPSC efficiency. The Li-TFSI contents regulated the amount of oxidized spiro-OMeTAD species, which was the central controller of device performance. Ultrafast spectroscopy revealed that hot holes and electrons were injected concomitantly and, consequently, no hysteresis was detected in the reversed scan of the I–V curve. Finally, the system was very transparent, with performances that showed it to be adequate for use as an energy harvester to power small devices, such as IoT sensors and e-paper displays.

## 2. Materials and Methods

The solid-state DPSC devices studied herein have the architecture shown in Figure 1a with the energy levels shown in Figure 1b, in accordance with what has been published elsewhere [27]. Briefly, the light absorbing material (Au nanoparticles (NPs)) was sandwiched between two transporting layers, an electron transporting layer (ETL), in this case TiO_2_, and a hole transporting layer (HTL), in this case spiro-OMeTAD. The ETL was deposited on a transparent substrate (FTO glass), and the cell was closed with a metal back contact.

The devices were prepared as follows. The FTO (TEC 15) substrates (NSG-Pilkington, UK) were cleaned with 1% Hellmanex solution (Sigma-Aldrich), water, acetone (Sigma-Aldrich) and isopropanol (Sigma-Aldrich) for 30 min, respectively, and UV/ozone-treated for 15 min (Ossila, UK). Afterwards, a thin layer of TiO_2_ was deposited via spray pyrolysis on a hot plate (500 °C) using a titanium diisopropoxide bis(acetylacetonate) (Sigma-Aldrich) precursor. The film was annealed for 30 min at 500 °C. The Au NPs were produced via a top-down approach. Briefly, a 3 nm Au film (Au grains from Metalor, Sweden) was thermally evaporated onto the TiO_2_/FTO substrate with a deposition rate of 0.1 Å/s. Subsequently, the film was annealed in air at 450 °C for 30 min to form Au NPs on the TiO_2_ surface. The spiro-OMeTAD (Sigma-Aldrich) hole transport layer was prepared by dissolving 18 mg of spiro-OMeTAD in 100 µL anhydrous chlorobenzene (Sigma-Aldrich). Two chemicals, 4-tert-butylpyridine (tBP) (Sigma-Aldrich) and bis(trifluoromethane)sulfonimide lithium salt (Li-TFSI) (Sigma-Aldrich)were added as additives and the molar ratio was 4:1. The Li-TFSI salt was dissolved in the anhydrous acetonitrile and added into the spiro-OMeTAD solution at different molar ratios. Around 25 µL of this solution was deposited onto a 2.5 × 1.5 cm^2^ substrate (Ossila spin coater) at 2000 rpm for 30 s. Then, samples were stored in a desiccator for 24 h and kept away from light. Finally, an 80 nm thick Au electrode was evaporated on top at a rate of between 0.1 and 0.8 Å per second.

Scanning electron microscopy (SEM) (Oberkochen, Germany) was performed using a Zeiss Leo 1530 SEM equipped with an in-lens detector.

UV-Vis absorbance was performed on Cary 5000 UV-Vis and NIR spectrophotometer (Santa Clara, CA, USA), and the baseline was collected from the FTO/TiO_2_ substrate. Reflectance (R) and transmission (T) measurements were taken by using the diffuse reflectance accessory. Therefore, the absorption was calculated according to the equation: Abs = 1 − R − T.

Raman spectroscopy measurements were carried out on a Renishaw Reflex Raman microscope (Järfälla, Sweden). The excitation wavelength was set to 532 nm, and a 50× objective was employed. Using a single-crystal Si reference, the system was calibrated with the strong phonon mode at 520.5 cm^−1^ using a 2400 lines/mm grating. The data were acquired from 300 scans for each measurement.

The current–voltage (I–V) characteristics of the devices were measured at room temperature using an Ossila Solar Cell I–V Test System (Sheffield, UK). The device performance was measured under AM 1.5G at 76.4 mW/cm^2^ (Pico LED solar simulator, G2V Optics), and a filter with a cut-off wavelength of 560 nm was used. The active area of each solar cell was 0.065 cm^2^ defined by the mask aperture during the measurement. The incident photon-to-electron conversion efficiency (IPCE) spectrum was measured by an instrument containing a xenon lamp (Spectra Products ASBXE 175), a potentiostat (PINE AFRDE 5), a LabJack U6 and a monochromator (Spectra Products CM110). The value of absorbed photon-to-electron conversion efficiency (APCE) was obtained by dividing the maximum IPCE by the peak of absorption.

Ultrafast transient absorption spectroscopy experiments were performed. The detection of ultrafast hot-hole injection into the spiro-OMeTAD valence band was monitored via transient infrared absorption spectroscopy (TIRAS). TIRAS experiments were carried out in a femtosecond transient absorption spectrometer (Helios IR, Ultrafast Systems LLC) (Sarasota, FL, USA) at room temperature. Briefly, the output of a Ti:sapphire amplifier with integrated oscillator and pump lasers (800 nm, 40 fs, 3 kHz, Libra LHE, Coherent) (Santa Clara, CA, USA)) was split into two beams, which were used to pump two TOPAS-Prime optical parametric amplifiers coupled with frequency mixers (Light Conversion). This setup produced a depolarized visible pump pulse (*λ*_pump_ = 550 nm, power 220 μW and fluency 1.3 × 1014 ph/pulse/cm^2^) and was broadly adjusted using a neutral density filter placed before the sample. Before reaching the sample, the probe beam was split into the equal-intensity probe and reference beams. Probe and reference beams were detected using a femtosecond transient absorption spectrometer. The instrument response function for the experiments was approximately ~150 fs.

The plasmon dynamics of the Au nanoparticles were monitored with femtosecond transient absorption spectroscopy (TAS) across the visible regime. TAS was performed on a TOPAS-Prime, Light Conversion (Vilnius, Lithuania). Briefly, a 40 fs pulsed laser with 3 kHz repetition rate and a fundamental laser pulse centered at 795 nm was used to generate pump pulse (*λ*_pump_ = 595 nm, power 500 μW and fluency 2.2 × 1015 ph/pulse/cm^2^) and probe light (*λ*_probe_ = 350 to 700 nm). Our instrument response function is ca. 150 fs, which was obtained by redirecting the pump and probe lights to a Newport MS260i spectro of (West North Logan, UT, USA). The kinetic traces were fitted with a sum of convoluted exponential functions.

## 3. Results and Discussion

The cell consisted of an FTO glass covered with ca. 35–55 nm of compact TiO_2_ layer, scattered Au nanoparticles (NPs) with an average size of 15 nm and ca. 620–680 nm layer of spiro-OMeTAD, as revealed by scanning electron microscopy (SEM) of the device cross-section presented in Figure 2a. The SEM top view of the Au NPs (Figure 2b) reveals well-dispersed particles that were homogeneous in size with a distribution of around 10%. The particles had a spherical shape (Figure 2a).

Figure 3a shows the UV-Vis spectra of Au NPs on TiO_2_ before and after deposition of the spiro-OMeTAD layer. Before adding the spiro-OMeTAD, the spectrum between 400–800 nm was dominated by the plasmon resonance of the Au NPs centered at 620 nm. The addition of spiro-OMeTAD led to a sharp feature below 430 nm, associated with the spiro-OMeTAD and a small shift in the plasmon resonance to higher wavenumbers, related to the change in dielectric medium from air to spiro-OMeTAD. It is worth noticing that the addition of spiro-OMeTAD reduced the delta absorbance between 430–580 nm, associated with Au NPs’ reflection and backscattering, furthering the transparency of the device. As DPSC are being developed to power IoT devices, transparency and color are important parameters for their adoption. Figure 3b shows that the device without the metal back contact was highly transparent with a faint blue hue, certifying the prospectus for developing photovoltaic energy-harnessing systems with significant aesthetic appeal. The current cells had been terminated with a gold back contact that was not transparent because terminations with transparent back contacts, such as ITO or AZO, produce inconsistent results. Therefore, to assess the effect of spiro-OMeTAD oxidation, we opted to use a gold back contact.

The compact TiO_2_ layer consisted of anatase, confirmed by the appearance of a sharp peak at 149 cm^−1^ on the Raman spectrum (not shown), ascribed to the E_g_ mode of anatase [28]. Figure 4 shows the Raman spectra of spiro-OMeTAD with different Li-TFSI amounts, which is the parameter that changes the amount of spiro^+^TFSI^−^ species that are responsible for spiro-OMeTAD hole-transporting properties [29]. The overall spectra matched what has previously been reported for spiro-OMeTAD [30] both in terms of peak position and relative intensity. Moreover, the spectra were reproducible throughout the layer based on Raman spectra collected on several parts of the films. Further analysis of the peaks in the region between 600–900 cm^−1^ was performed to establish the extent of oxidized spiro in the sample (this is discussed later in the text).

Figure 5a shows a representative I–V curve of the solar cell’s devices, in this case, a device with spiro-OMeTAD with 20% Li-TFSI. The sample was illuminated with a solar simulator with a filter that removed light below 560 nm, thus isolating the plasmon contribution to the photocurrent from any other potential sources, such as the photocurrent from the transporting layers that fall below 450 nm. The I–V curve shows characteristic diode behavior without any hysteresis. From the curve, one could estimate the open circuit voltage (*V*_oc_), short-circuit photocurrent (*J*_sc_) and the fill factor, which we found to be 0.23 V, 65 μA/cm^2^ and 31%, respectively. Figure 5b shows that the incident photon-to-electron conversion efficiency (IPCE) curve followed the absorption of the Au LSPR band, confirming that the measured values related to charging produced by Au plasmon excitation, not intra- or inter-band excitation. From the IPCE, one can estimate the absorbed photon-to-current conversion efficiency (APCE), calculated to be around 4.2%. The value is within what Reineck et al. measured for a solar cell with Au particles of around 15 nm [23].

TIRAS measurements were performed to confirm charge injection into the transporting layer. Free carriers have strong absorption in the infrared region, yielding a characteristically broad and featureless positive absorption [18,31]. Figure 6a shows the kinetic trace extracted at 3605 nm and depicts the temporal behavior of the free carriers injected into the transporting layers. Plasmons can inject charge directly (coherent injection) or indirectly via hot carriers [32]. The present device uses accepting layers that have no optical overlap with the plasmon, thus not in direct injection conditions. Therefore, the prevalent mechanism should be hot carrier injection, which in Au is expected ca. 100 fs after plasmon excitation [10].

Starting with the Au supported on TiO_2_, its excitation at 550 nm led to a rapid increase in signal intensity, with a rising edge shorter than our instrument response function (≈150 fs), suggesting that electrons are injected into TiO_2_ in less than 150 fs. An injection time faster than the instrument resolution was also detected for the hole injection into spiro-OMeTAD from Au in the absence of TiO_2_. The addition of spiro-OMeTAD resulted in a significant intensity increase, suggesting that electrons and holes were being injected. As observed with Hattori et al.’s [14] system (TiO_2_/Au/PEDOT:PSS), the signal was significantly higher than the sum of the individual components, suggesting an improvement in charge separation when both transporting layers are present. Another important observation is that the addition of spiro-OMeTAD did not affect the rising edge component, which is better observed in Figure 6b; this observation suggests that holes are injected within 150 fs, similar to hot electrons. As hot carriers are expected to take about 100 fs to be formed in Au [10], and the injection occurs within 150 fs despite the differences in the charges’ effective mass and tunnelling distances, it is reasonable to assume that they are injected concurrently. Until now, this has been an open question. Reineck et al. [22] proposed three potential scenarios behind the injection of hot carriers (Figure 7), suggesting an electron and hole injection occurring concurrently or one after the other. Based on the TIRAS experiments, one can now state that scenario C is the most likely, which means no significant potential is built at the Au particle due to temporal differences in electron and hole injection. This is also consentient with the lack of hysteresis in the I–V curve and with previous measurements with (TiO_2_/Au/PEDOT: PSS) [14]. It is expected that the transporting layer thickness’ affects primarily charge separation and recombination kinetics, not charge injection.

A detailed analysis of TIRAS decay is out of the scope of the manuscript, but it is clear that most charge decays within 1 ns, which limits the charge collection at the electrodes and, consequently, leads to relatively low photocurrents, as observed elsewhere [33]. Charge extraction at the electrodes is significantly suppressed for devices where charge recombines within less than 1 ns, a commonly used rule of thumb to correlate transient data with steady-state performance. Furthermore, the addition of spiro-OMeTAD introduced the possibility for interfacial recombination between the transporting layers apart from back-transfer to the Au plasmon, which is visible by the presence of new decay functions.

Having settled the most likely mechanism behind Au plasmon hot carrier injection into the transporting layers, the performance of the solar cells with different Li-TFSI content was tested, as shown in Figure 8. The reported performance values relate to the average of at least eight devices. It is clear from data analysis that the performance peaked when the Li-TFSI content was 20%. It should be mentioned that performance was measured within 12 h after adding the spiro-OMeTAD, which is important for understanding the behavior, as discussed below. Starting with *V*_oc_, the values ranged from 200–240 mV, consistent with what has been reported elsewhere [22]. The relatively low *V*_oc_ can be partially justified by the extremely thin ETL layer leading to pinning of the electronic levels. To confirm this, a device with a thicker ETL was made. The device had a *V*_oc_ close to 500 mV, but a significantly lower *J*_sc_, justified by the significant increase in the ETHL layer resistance. Note that ETL optimization is out of the scope of the study. The *J*_sc_ also showed a maximum for devices with 20% Li-TFSI content, which will be explained later. Consequently, and unsurprisingly, the power production scatterplot is similar to the other performance values with a maximum for devices with 20% Li-TFSI content.

A potential hypothesis for the observed differences is that the Li-TFSI content in the spiro-OMeTAD changes the capabilities of HTL hole extraction. To evaluate this, we determined the electron–phonon (e–ph) lifetime upon Au plasmon excitation at 595 nm via transient absorption spectroscopy (TAS). This parameter was recently shown to be very sensitive to the number of holes extracted from Au [18]. The TiO_2_ layer was replaced by ZrO_2_, which has similar properties but a significantly larger bandgap impeding the electron injection to avoid interference from the electron injection. Plasmon excitation broadened its absorption, yielding a characteristic pump-probe signal consisting of bleach around the pumping wavelength and two positive winglets on the side of the bleach signal (see Figure 9) [34,35,36,37,38]. By fitting the transient signal, one can extract the electron–electron (e–e), electron–phonon (e–ph) and phonon–phonon (ph–ph) lifetimes.

From the kinetic traces presented in Figure 10, the e–e lifetime could be extracted from the rising edged function of the signal, which was found to be around 150–200 fs for all systems. The e–ph lifetime was extracted from the first decay function of the signal. The extracted values are summarized in Table 1. The table shows that the values were similar, independent of where they were extracted and the Li-TFSI content. The observation suggests that Li-TFSI content does not affect the hole injection and thus cannot be responsible for the observed changes in DPSC performance.

The second hypothesis for the observed changes in DPSC performance relates to the concentration of oxidized spiro-OMeTAD (spiro^+^TFSI^−^). Li-TFSI is commonly added to spiro-OMeTAD to regulate the amount of spiro^+^TFSI^−^, ascribed as the species responsible for hole conductivity on this HTL. Raman has been found to be sensitive to this species, and thus it is a good indicator of its relative abundance. More specifically, Lamberti et al. assigned the peak at 731 cm^−1^ to the C-S group in the spiro^+^TFSI^−^ [39]. Its natural abundance in the samples can be estimated by comparing it with the intensity of a Li-TFSI peak at 760 cm^−1^ (Figure 11). There is noticeably a peak at around 710 cm^−1^ associated to tBP, as reported elsewhere [40].

Table 2 summarizes the analysis outcome. From this, one can observe a good correlation between our DPSC device performance and the relative content of spiro^+^TFSI^−^, suggesting this to be an important parameter to consider when trying to optimize the performance of the solar cell. According to Schölin et al. [29], the addition of Li-TFSI to the spiro-OMeTAD affects the Fermi level energy and valence band position. However, the changes reach a saturation level after the addition of 5% of Li-TFSI. We purposely selected Li-TFSI concentrations that would ensure that these parameters were constant, which is also consistent with some preliminary UPS measurements. We did not detect significant differences in the spiro-OMeTAD film’s conductivity measurements with different Li-TFSI concentrations, suggesting that the spiro-OMeTAD layer thickness masks the effect of Li-TFSI on film conductivity. Therefore, the local Raman analysis of species concentration provided a better insight into the role of spiro^+^TFSI^−^ on the cells’ efficiency, especially considering the film’s relative homogeneity based on Raman map analysis.

The amount of spiro^+^TFSI^−^ on the samples was expected to follow the Li-TFSI content, which is not consistent with our measurements. However, we would like to remind the reader that characterization and other tests on the samples were performed within 12 h of sample preparation, which might not be sufficient to achieve steady-state levels of oxidation. This was partially validated when we measured the UV-Vis spectra of spiro-OMeTAD layers left in the dark for 1 month. Longer oxidation times yielded the characteristic UV-Vis absorption band and a significant change in the visual color of the samples (Figure 12). A study of this effect is out of the scope of the work but helps to understand the observed discrepancy.

## 4. Conclusions

Herein, the effect of Li-TFSI content in HTL on DPSC performance was studied. The findings revealed that the amount of oxidized spiro-OMeTAD (spiro^+^TFSI^−^) significantly conditioned device performance. The optimal performance was obtained on a device containing a spiro^+^TFSI^−^ relative abundance of around 33%. The Li-TFSI content did not affect charge extraction, as revealed by analysis of electron–phonon lifetime. TIRAS experiments suggested that both hot carriers are injected simultaneously and shortly after their formation, which was corroborated by the lack of hysteresis in the I–V curves. The spiro-OMeTAD layer reduced Au nanoparticles’ reflection/backscattering, which improved the overall cell transparency. The fabricated DPSC can be made highly transparent, and their performance was found to be within what is required for low-power devices, such as IoT sensors and e-paper displays, according to Haight et al. [4]. Further improvements in cell performance can be achieved by optimizing charge transporting layer thickness and suppressing interfacial recombination.

## Figures and Tables

**Figure 1 nanomaterials-11-03329-f001:**
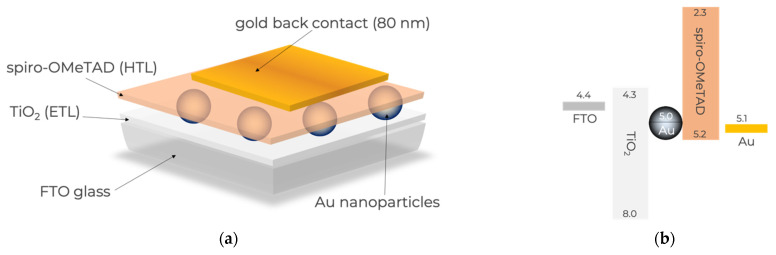
Schematic representation of n–i–p DPSC architecture. (**a**) Constituents of the DPSC used herein; (**b**) energy level diagram of the main constituents according to what has been published elsewhere [26].

**Figure 2 nanomaterials-11-03329-f002:**
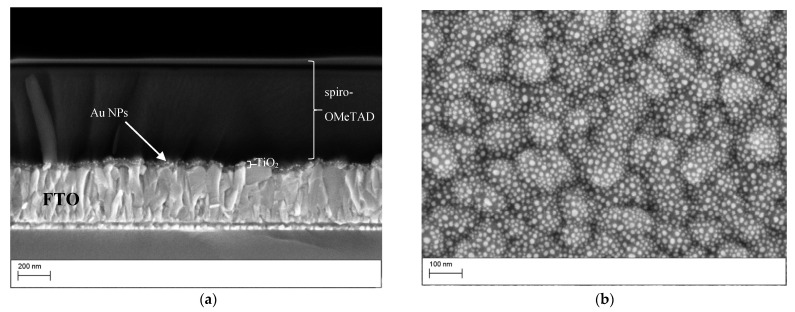
Scanning electron microscopy (SEM) analysis. (**a**) Cross-section SEM of a representative DPSC consisting of FTO/TiO_2_/AuNPs/spiro-OMeTAD; (**b**) SEM of Au NPs on TiO_2_/FTO created from annealing a 3 nm evaporated Au layer at 450 °C.

**Figure 3 nanomaterials-11-03329-f003:**
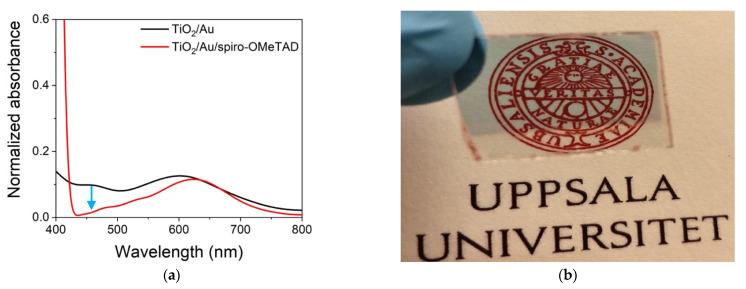
Optical characteristics of the DPSC. (**a**) UV-Vis spectra of the device before and after adding spiro-OMeTAD. The spectra were the subtracted by the FTO/TiO_2_ film absorbance to highlight the differences; (**b**) optical photograph of a representative DPSC before evaporation of gold back contact.

**Figure 4 nanomaterials-11-03329-f004:**
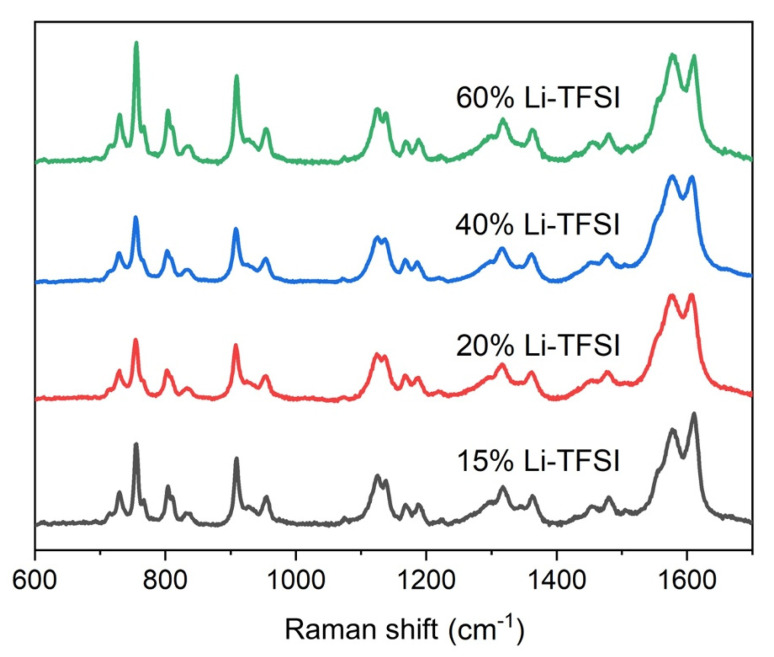
Raman spectra of spiro-OMeTAD with different Li-TFSI amounts.

**Figure 5 nanomaterials-11-03329-f005:**
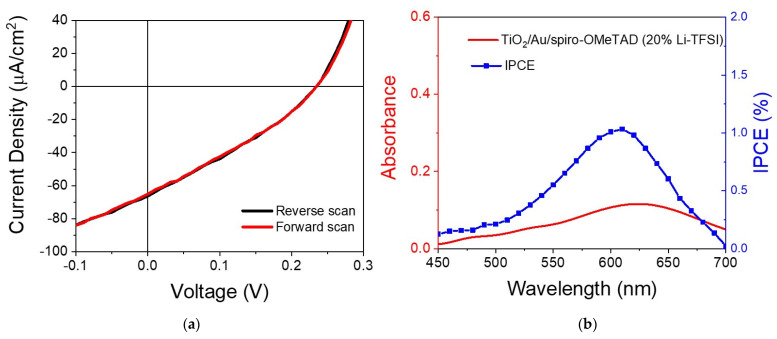
(**a**) I–V curve of the device containing spiro-OMeTAD with 20% Li-TFSI measured under simulated solar conditions with a cut-off filter that removes any light below 560 nm; (**b**) IPCE curve versus optical absorbance of the device.

**Figure 6 nanomaterials-11-03329-f006:**
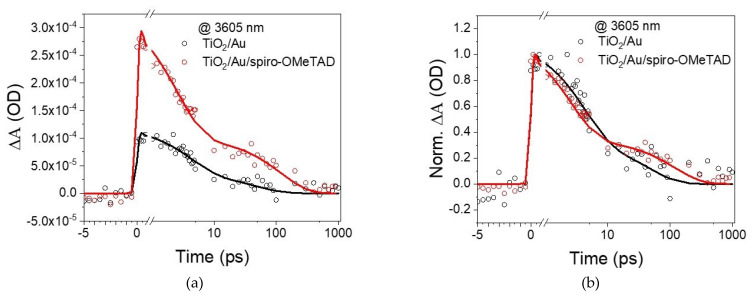
TIRAS measurements depicting charge injection for device containing spiro-OMeTAD with 20% Li-TFSI upon Au plasmon excitation at 550 nm. (**a**) Effect of adding the spiro-OMeTAD on the intensity of the transient data extracted at 3605 nm; (**b**) normalized intensity curves depicting the effect of adding the spiro-OMeTAD on the temporal evolution of the signal.

**Figure 7 nanomaterials-11-03329-f007:**
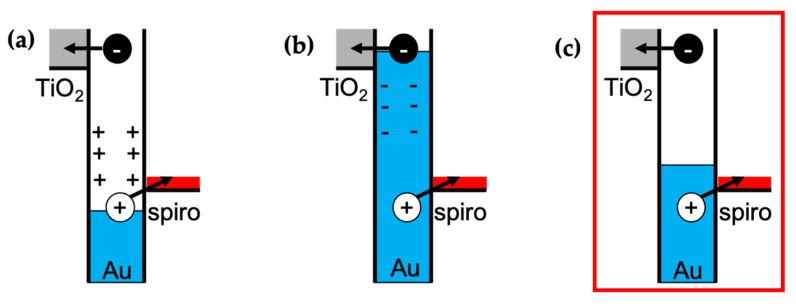
Schematic representation of the possible injection mechanism upon plasmon excitation. The red square highlights the mechanism consistent with the TIRAS data. (**a**) Electron injection followed by hole injection; (**b**) hole injection followed by electron injection; (**c**) simultaneous injection of electron and hole. The red square indicates the best fitting injection mechanism based on TIRAS measurements.

**Figure 8 nanomaterials-11-03329-f008:**
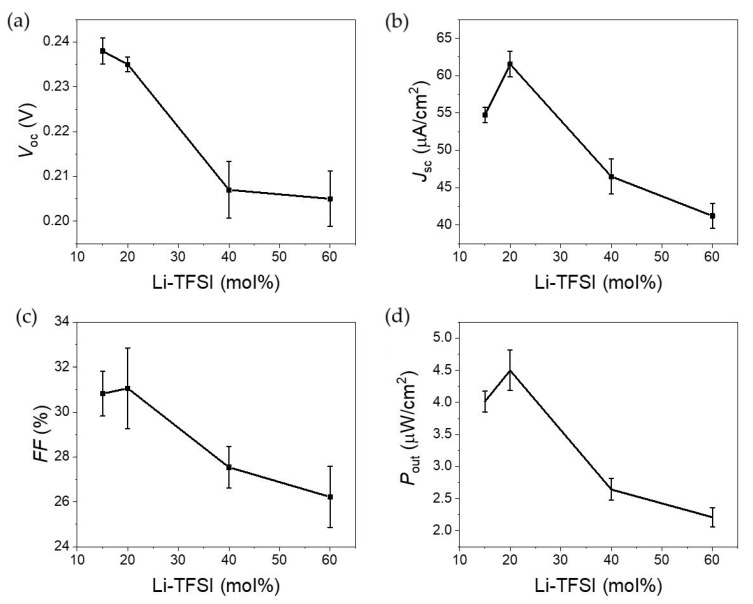
DPSC device performance as a function of Li-TFSI content spiro-OMeTAD under solar simulate illumination with a cut-off filter removing light below 560 nm. (**a**) Open-circuit voltage (*V*_oc_); (**b**) short-circuit photocurrent (*J*_sc_); (**c**) fill factor in percentage (*FF*); (**d**) power output (*P*_out_ = *V*_oc_ × *J*_sc_ × *FF*).

**Figure 9 nanomaterials-11-03329-f009:**
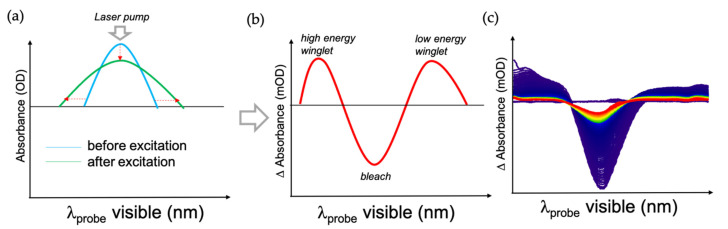
TAS measurement of plasmonic signal. (**a**) Schematic representation of the change in plasmonic absorption due to optical excitation; (**b**) different spectra measured in pump-probe; (**c**) example of data collected as a function of time (purple: shorter delay times, red: longer delay times).

**Figure 10 nanomaterials-11-03329-f010:**
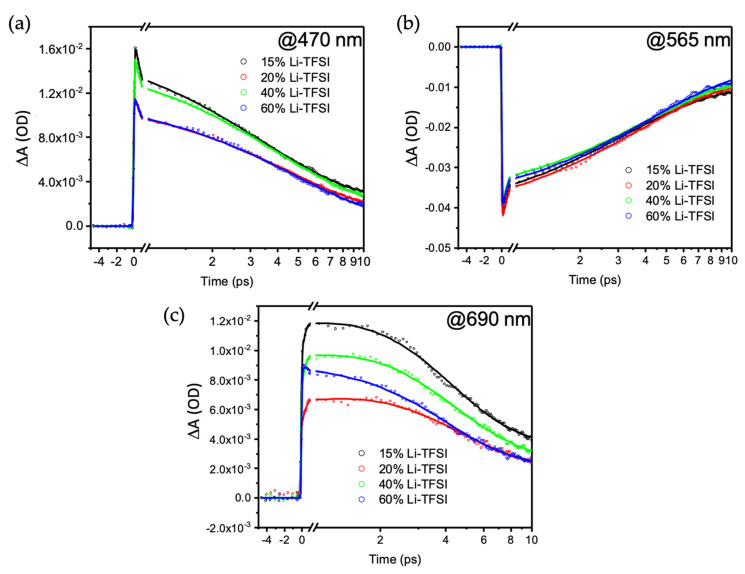
Kinetic traces extracted from the TAS measurement upon Au plasmon excitation at 595 nm. (**a**) Signal extracted at 470 nm (winglet high-energy side); (**b**) signal extracted at 565 nm (bleach); (**c**) signal extracted at 690 nm (winglet low-energy side).

**Figure 11 nanomaterials-11-03329-f011:**
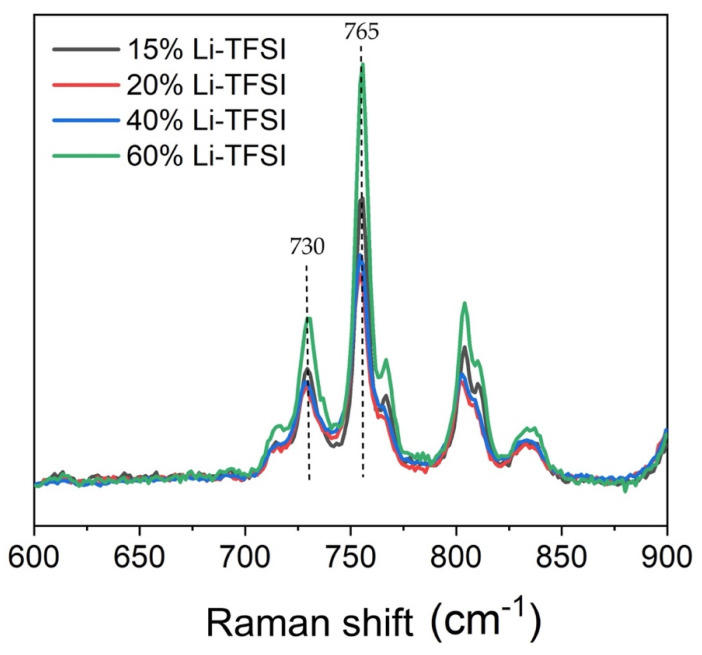
Raman spectra of C-S group in the spiro^+^TFSI^−^ species and neutral spiro-OMeTAD with different Li-TFSI amounts.

**Figure 12 nanomaterials-11-03329-f012:**
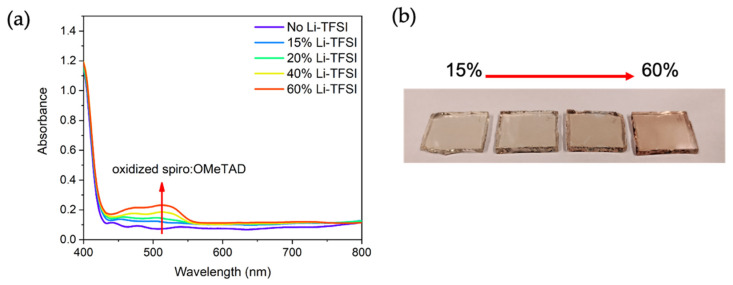
Optical properties of oxidized spiro-OMeTAD layers after 1 month in a desiccator in the dark. (**a**) UV-Vis spectra with different Li-TFSI contact; (**b**) optical picture.

**Table 1 nanomaterials-11-03329-t001:** Optical changes in the Au LSPR absorption maximum due to changes in spiro-OMeTAD conductivity caused by changes in Li-TFSI concentration.

Wavelength (nm)	15% Li-TFSI (ps)	20% Li-TFSI (ps)	40% Li-TFSI (ps)	60% Li-TFSI (ps)
470	3.4 ± 0.1	4.5 ± 0.2	4.0 ± 0.1	4.8 ± 0.3
565	3.2 ± 0.1	3.8 ± 0.2	4.0 ± 0.2	4.2 ± 0.2
690	3.8 ± 1.5	4.1 ± 1.7	4.1 ± 0.6	3.8 ± 0.2

**Table 2 nanomaterials-11-03329-t002:** Raman intensity analysis to quantify abundance of oxidized spiro-OMeTAD in the cells.

Raman Peak	15% Li-TFSI	20% Li-TFSI	40% Li-TFSI	60% Li-TFSI
A (Intensity @ 730 cm^−1^)	1.19	1.04	1.05	1.74
B (Intensity @ 765 cm^−1^)	3.02	2.11	2.41	4.47
A/(A + B)	0.28	0.33	0.30	0.28

## Data Availability

Not applicable.

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
