# Peer review of "Direct Plasmonic Solar Cell Efficiency Dependence on Spiro-OMeTAD Li-TFSI Content"

_nanomaterials, 2021, doi:10.3390/nano11123329_

Round 1
Reviewer 1 Report
Overall this is an orticle fo a good standard for publication. I would recommend the following changes before publication.
Abstract: Justify or remove the statement that the amount of oxidized Spiro is the primary regulator of devices performance. I agree it does regulate the performance but it will not be the only thing that does impact it significantly so this reads as an overstatement. line 445 where it is descried as a critical parameter is a better choice of words.
A more detailed description of how direct plasmonic soalr cells should be added at the end of the paragrpah starting line 68. What dirves the transport of the hot carriers the transport layers?
Sentence starting on Line 79 needs further clarification - by not compact do you mean it is made of nanoparticles? What is meant by to prevent interference on the plamon resonance?
Figure 1 . The authors should consider adding an energy level diagram to figure 1 or referrnging to figure 7 at this point in the text to explain how these devices function
line 151 there is a section heading / incomplete heading at the start of this paragraph, but no headings elsewhere.
line 178 - the statement that the NPs are highly homogenous is not well justified by the data fig 2(b) suggests there is a range in size - please quantify this. Also commment on the shape of the NPs. Spherical or hemispherical?
There are repeated references to semitransparent devices, but with a gold back contact this will not be the case. This need clarification.
There are frequent references to the Au layer - this needs to be clarified throughout as there are 2 Au layers, the Au NPs and the back contact. eg. line 301, 311
line 229, elaorate on the other potential sources
The devices are low power, low efficiency, comment in discussion on how useful this would be for IoT devices compared to a smaller area of Si PV.
Figure 5 why not use all the area for data, change max on y axes to Abs-0.2 and IPCE=1.2% Then it would be apparent that the peaks are not very well matched.
Figure 7 - the caption needs to explain what the blue, grey and red areas represent. Do you mean (c) when you refer to the red square in caption?
Comment on the number of repeat devices used to generate the data in figure 8.
Line 369 clarify what you mean by volcano plot
Figure 10 caption was obscured behind the figure in the version I got
It's not possible to see the number/positions of data points on teh rising edge so difficult to determine if the fit is reasonable. line 415
Conclusions: same comment on oxidised spiro being a critical parameter, but not necessarily the main regulator (it may be the main regulator of those you considered)
Line 480 Remove the comment about highly transparent devices or explain how they are so with a Au back contact
The English could be improved in a few places throughout the document.
Author Response
Pl

Reviewer 2 Report
In this work, the authors studied the effect of Li-TFSI doping on Spiro-2 MeTAD and tried to understand the photophysical and optoelectronic properties in direct plasmonic solar cell (DPSC). This work will provide the interest of photovoltaic and photophysics community. However, there are some concerns which should be addressed before publishing the work in “Nanomaterials”.
- Although the instrument response of TAS setup was mentioned, there is no information about pump laser excitation intensity. Please add laser fluence used in ultrafast measurement.
- Authors tried to connect ultrafast charge recombination with steady state photocurrent measurement. Please add some references relevant to that or add time delay collection field (TDCF) experiments to justify that. They mentioned that (page 7) “it is 320 clear that most charge decays within 1 ns, which justifies the relatively low photocurrent”.
- In page 9, authors mentioned “The relatively low Voc can be partially 366 justified by the extremely thin ETL layer”. Please elaborate it and add some references.
Author Response
file attached

Reviewer 3 Report
The authors have studied the hole conductor effects (LiTFST content) on the performance of plasmonic solar cells, They found that LiTFST content did not influence the charge injection, but modulate the film transparency. However, there are some concerns regarding the experimental results and the discussion. The performance enhancement relating to the LiTFSI content is not clear. Overall, this work show insignificant novelty and some inconsistent results.
- In the introduction, the author stated that the potential of such DPSC in IoT application, but how much power required in such devices and whether such low performance are feasible for IoT? More explanation about it will be helpful.
- The direct contact of TiO2 and spiro could induce severe charge recombination because of uncompact Au nanoparticles? How can the authors explain that?
- Very low power conversion efficiency is obtained for such device, has the authors measured the device (TiO2/spiro) performance without Au nanoparticles? Previous work has demonstrated much higher performance for the device without any light harvesting layer (J. Mater. Chem. A, 2020,8, 15670-15674).
- The authors measured the Raman spectra of spiro-LiTFSI, but more clear explanation regarding the effects of LiTFSI on the peak intensity.
- If the conductance of Spiro is enhanced due to more LiTFSI, what are the reasons for the decrease of Voc, Jsc and FF for higher LiTFSI content? The solar cell performance related to LiTFSI content is confusing and now well explained. More results and discussion are required.
- For the device performance, how much light absorbing contribution from spiro rather than Au nanoparticle? Can spiro itself absorb light and convert to charge carriers? The device performance of FTO/TiO2/spiro/Au should be compared as a reference.
- What are the main limitations of this device performance? More discussion regarding device parameters evolution with LiTFSI content is needed.
Author Response
file attached

Round 2
Reviewer 3 Report
The authors have tried to answer the questions, but most of them are not well discussed and presented. Some comments are followed.
1. As mentioned by the authors that this work demonstrated to show the spiro+ content effects on device performance, then in the introduction more discussion should be added in terms of spiro composition composition rather than a long history of whole DPSC. The specific issues or questions regarding this work should be presented in the introduction. DSPC is a relatively new technology with many pertinent open questions. For example, as mentioned: Two pressing questions are: 1) are the two charges injected simultaneously or singly?; and 2) is the oxidize content of spiro:OMeTAD determinant for device performance? In this case, the backgroud regarding these two problems should be reviewed and discussed in the introduction.?
- As agreed by the authors, the direct contact between TiO2 and Spiro could induce recombination, I believe that this recombination will influence the electron injection as well hole injection kinetics, which is one of the key questions the authors like to discuss. How does this interface recombination interfere with your conclusion in "are the two charges injected simultaneously or singly"?
- If the oxidized spiro content has impacts on the device performance, the authors are suggested to discuss more about the reasons why more oxidized spiro promtes power conversion? how about conductivity, energy levels and film quality?
Author Response
please find file attached
